# Outcomes of Combined Anterior Cruciate Ligament and Anterolateral Ligament Reconstruction According to GNRB Arthrometer Measurement

**DOI:** 10.3390/medicina59020366

**Published:** 2023-02-14

**Authors:** You-Hung Cheng, Chih-Hao Chiu, Alvin Chao-Yu Chen, Yi-Sheng Chan, Kuo-Yao Hsu

**Affiliations:** 1Department of Orthopedic Surgery, New Taipei Municipal Tu-Cheng Hospital, Chang Gung Memorial Hospital, New Taipei City 236, Taiwan; 2Comprehensive Sports Medicine Center, Linkou Chang Gung Memorial Hospital, Taoyuan City 333, Taiwan; 3Department of Orthopedic Surgery, Linkou Chang Gung Memorial Hospital, Taoyuan City 333, Taiwan

**Keywords:** ACL, ALL, GNRB arthrometer

## Abstract

*Background and Objectives*: To investigate the prognosis of combined anterior cruciate ligament (ACL) and anterolateral ligament (ALL) reconstruction, we used a GNRB (Genourob, Laval, France) arthrometer to measure surgical outcomes. *Materials and Methods*: This retrospective study reviewed patients who underwent combined ACL and ALL reconstruction and had a minimum follow-up of 2 years. Subjective outcomes, namely the International Knee Documentation Committee (IKDC) evaluation form scale scores and Lysholm scores, were evaluated preoperatively and postoperatively. We used a GNRB arthrometer to test the side-to-side laxity under pressures of 134 and 200 N, and we calculated the differential of the slope of the curves. We also recorded complications. *Results*: Our study examined 18 patients (mean age: 30.56 ± 8.9 years, range: 19–53) with a mean follow-up of 27.37 ± 3.4 months (range: 24–36). Both Lysholm and IKDC scores were significantly improved following the operation. The GNRB arthrometer measured mean anteroposterior laxity side-to-side as 0.76 ± 0.78 mm and 0.82 ± 0.8 mm under pressures of 134 and 200 N, respectively. The mean side-to-side differential slope under 200 N was 3.52 ± 2.17 μm/N. These values indicated that patients displayed no graft tear or low functional knee instability. All patients had a grade 3 pivot shift preoperatively; only two patients had a grade 1 pivot shift postoperatively, with the rest having a negative pivot shift. *Conclusions*: Our study revealed that combined ACL and ALL reconstruction has an excellent prognosis. GNRB measurement demonstrated excellent stability, and most patients had no residual pivot shift.

## 1. Introduction

Anterior cruciate ligament reconstruction (ACLR) is a well-accepted treatment with satisfactory results for anterior cruciate ligament (ACL) injuries. The primary goal of ACLR is to restore the anterior and rotatory stability of the knee; however, graft failure is still reported in more than 10% of cases, and residual instability remains a problem after ACLR [1,2,3]. Studies have identified the importance of the anterolateral region over the knee and focused on the anatomy of the anterolateral ligament (ALL) [4,5]. Research has investigated the biomechanics of the ALL and demonstrated its role in resisting internal rotation, especially when the knee is in flexion [6,7,8,9,10]. Several surgical procedures have been developed for augmentation near the anterolateral corner since synergistic stabilization with ALL was introduced, and one of the most popular methods is ALL reconstruction [11,12,13,14]. A systemic review indicated that combined ACL and ALL reconstruction might improve pivot shift and increase knee rotatory stability, and it may have lower graft re-rupture rates [15]. The clinical and functional outcomes do not seem inferior to isolated ACL reconstruction. However, ALL reconstruction’s beneficial prognosis has primarily been studied on the basis of the subjective functional outcome rather than objective measurement.

Traditionally, several arthrometers were available to quantify anterior tibial translation. The KT-1000 (Medmetric, San Diego, CA, USA) and the Rolimeter (Aircast, Summit, NJ, USA) were the most frequently used. However, both devices were operator-dependent. The Telos (Telos GmbH, Laubscher, Hölstein, Switzerland) was another widely-used device which produced knee stress for radiographic images. Although it is reproducible, exposure to radiation was considered a disadvantage to patients. The GNRB (Genourob, Laval, France) arthrometer was introduced in 2007. The GNRB’s automated dynamic pressure and movement sensors can yield more accurate measurements than prior methods. In the field of ACL analysis, the GNRB arthrometer is considered more reproducible and user-friendly. Using the automated dynamic tests, the GNRB can collect data regarding patients’ knees that are more precise. Collette et al. reported both superior intra-examiner and inter-examiner reproducibility of the GNRB over the KT-1000 [16]. Another study reported that the GNRB is more effective than the Telos for detecting partial or complete ACL tears [17]. Researchers have reported some prognosis findings regarding ACLR using the GNRB; however, few studies have focused on ALL reconstruction. Therefore, we aimed to evaluate the objective outcome of combined ACL and ALL reconstruction using our institute’s GNRB arthrometer for measurement.

## 2. Materials and Methods

### 2.1. Study Design

This study was a retrospective cohort review. We enrolled patients with an ACL tear who underwent combined ACL and ALL reconstruction at our institute at any time between January 2018 and September 2019. The criteria for simultaneous reconstruction of both ligaments were a grade 3 pivot shift under physical examination, chronic ACL lesion, high level of sporting activity, or radiograph imaging indicating ALL injury (lateral femoral notch sign or Segond fracture). We excluded patients with less than 2 years of follow-up and patients who refused to participate in the study.

Using our institution’s electronic medical records, we reviewed patients’ charts for general data, radiographs, and magnetic resonance imaging. We reviewed patients’ International Knee Documentation Committee (IKDC) evaluation form scale scores and Lysholm scores preoperatively and postoperatively. The clinic regularly performed and recorded physical examinations, including testing the range of motion and conducting a complete ligament examination.

### 2.2. Instrumented Testing Using a GNRB Arthrometer

GNRB measurements were conducted preoperatively and with postoperative follow-up after two years in the clinic. According to the manufacturer’s guidelines, patients were lying on a standard examination table in the supine position. Each knee was placed separately at 20° of flexion and 0° of rotation, and the healthy knee was examined first. After marking the patella and tibial tuberosity, the GNRB devices (Figure 1) were in-stalled, and the buckles were fastened. The minimum pressure over the patella was 60 N. A sensor was positioned at the marked tibial tuberosity and fixed. The GNRB was manipulated, with the measuring sequences being one push at 134 N, one push at 150 N, and three pushes at 200 N, progressively. The international reference force for assessing the ACL based on the KT1000 arthrometer is 134 N. After measurement, we selected 134 N push data and the most similar pressure obtained from each knee under 200 N for comparison. Two curves were then generated on a chart representing each knee. The GNRB provided an automatic calculation of differential laxity and the differential of the slope of the curves, which reflected ligament elasticity. All data were collected on a computer in the examination room. An experienced physiotherapist and orthopedic surgeon (Y-H.C.) used the GNRB and reviewed the data.

### 2.3. Statistical Methods

Descriptive statistics were used to quantify overall results. A paired *t*-test was used for independent samples to evaluate all preoperative and postoperative data; *p* values of <0.05 were considered statistically significant. All statistical analysis was performed using SPSS, version 20 (IBM, Armonk, NY, USA).

### 2.4. Surgical Technique

All surgeries were performed by a single surgeon (K-Y.H.). After a patient was anesthetized, they were placed in a supine position with a tourniquet. Bony landmarks of the ACL and ALL (the patella, tibial tuberosity, fibular head, and Gerdy’s tubercle) were marked before surgery. Hamstring tendons, including semitendinosus and gracilis autografts, were first harvested and prepared for ACL and ALL grafts. Two holes were then created and connected near the anterolateral knee for passage of the ALL graft. Arthroscopic surgery was then performed using standard portals for ACL reconstruction. After the ACL femoral and tibial tunnels were created, the combined ACL and ALL grafts were passed from the tibial tunnel to the femoral tunnel and then fixed with bioabsorbable interference screws (Conmed). The additional strand of gracilis graft for the ALL graft was brought to the tibial region around Gerdy’s tubercle underneath the IT band from the femoral tunnel. Another interference screw was used to fix the ALL over the posterior hole. If the residual graft was long enough, the graft wound was once again pulled back to the femoral site under the IT band and tied with three sets of No. 5 Ethibond sutures over the end of the ACL graft (Figure 2 and Figure 3). The position of fixation was the knee in full extension and neutral rotation. Physical examination for stability, namely range of motion and Lachman and pivot shifts, occurred after the procedure. After confirming good stability, the wound was closed layer by layer.

## 3. Results

### 3.1. General Data

Between January 2018 and September 2019, 18 patients (14 men and 4 women) satisfied our inclusion criteria. The mean age at surgery was 30.56 years (range, 19–53 years), with a mean time from injury to surgery of 22.83 months. The mean duration of follow-up was 27.37 months (range, 24–36 months). Data and associated injuries are displayed in Table 1. We observed no major complications, such as neurovascular injury or wound infection, during or after surgery.

### 3.2. Subjective Knee Evaluation

The mean IKDC subjective scores were 54.3 ± 9.0 preoperatively and 85.1 ± 1.6 at the last postoperative follow-up. The mean Lysholm scores were 52.3 ± 3.3 before surgery and 98.1 ± 2.2 at the last follow-up. Both scores indicated significant improvement post operation. All patients had a preoperative grade 3 pivot shift test result. After the operation, 16 of 18 patients (88.9%) were negative for pivot shift, and the others had only a grade 1 pivot shift test result in the clinical follow-up (Table 2).

### 3.3. Objective Value of the GNRB Arthrometer

Using the GNRB arthrometer, the mean anteroposterior laxity side-to-side was 0.76 ± 0.78 mm under 134 N and 0.82 ± 0.8 mm under 200 N, indicating no ligament tear. The mean side to side differential (SSD) slope under 200 N was 3.52 ± 2.17 um/N, and a value below 5 um/N represented low functional instability (Table 3). All patients achieved a full range of motion in the final follow-up.

## 4. Discussion

Our study demonstrated good prognosis and objective value for combined anatomic ACL and ALL reconstruction. This study is the first to use the GNRB arthrometer to evaluate outcomes of ACL and ALL reconstruction and revealed that ACL and ALL reconstruction yielded excellent SSD anterior laxity and stability.

ACL reconstruction is a well-developed surgery that uses the minimally invasive arthroscopic technique to restore the anterior translation of the tibia and the rotatory instability caused by an ACL tear. Recently, the concept of “anatomic reconstruction” has achieved excellent results [18,19,20]. Even though the results of ACLR are satisfactory and reliable over time, up to 26.5% of patients still experience residual pivot shift after ACL reconstruction [2,3]. Therefore, researchers have analyzed treatment options that are performed around the anterolateral knee, of which two primary approaches exist. One is lateral extra-articular tenodesis, which has various approaches and modifications that have generated promising results [21,22]. The other is the simultaneous anatomic reconstruction of the ACL and ALL, which has given rise to more detailed examinations of ALL anatomy. Several variable surgical techniques have been developed with promising short-term prognoses [1,23,24]. However, some systemic reviews revealed variation in the anatomic footprint and heterogeneous biomechanical results, and the objective measurement of the surgical outcome also varied [25,26]. DePhillipo et al. found there was more variability in the femoral tunnel location than in the tibial tunnel. Most studies placed femoral tunnel posterior and proximal to the lateral epicondyle, whereas two studies reported a distal tunnel location [26]. Variation in surgical methods might lead to the bias for further objective measurement. In this study, we used anatomic ACL and ALL reconstruction introduced by Sonnery-Cottet et al. [23]. The grafts were passed from the tibial tunnel to the femoral tunnel, and the additional strand of gracilis graft for the ALL was brought to the tibial region between the Gerdy’s tubercle and the fibular head from the femoral tunnel, which was made near the lateral epicondyle posteriorly and proximally. ALL footprints have been demonstrated using cadaveric study and were well accepted [4,5]. A looping technique was used for augmentation over the anterolateral knee using the gracilis tendon as an autograft. Long-term results reported by the SANTI group indicated this technique had better ACL graft survivorship, lower overall rates of reoperation, and no increasing complications in combined ACL and ALL reconstruction compared with an isolated ACL tear [27]. A study used a Rolimeter for instrumented measurement of anteroposterior laxity, and the side-to-side difference averaged 0.7 ± 0.8 mm after combined ACL and ALL reconstruction, with a minimum 2-year follow-up [23]. The GNRB, the automated dynamic arthrometer in our study, measured the mean side-to-side differential anterior laxity as 0.76 ± 0.78 mm. Although knee laxity devices used in regular practice might not be comparable to those used in other studies [28], our tibial translation value was not inferior to that of SANTI’s group. Our results confirmed that simultaneous anatomic reconstruction of the ACL and ALL could achieve excellent results.

Accurately measuring anterior knee translation is crucial to preoperatively evaluating ACL/ALL reconstruction surgery’s laxity and prognosis. Numerous arthrometers, including the KT1000 and the Rolimeter, have been introduced to evaluate knee stability by manually recording the translation of the tibia. However, their reproducibility and reliability have been found to be poor or dubious [29,30]. The GNRB is an automatic arthrometer that uses dynamic force to minimize the error produced by the examiner. Earlier results have indicated that the GNRB has better interobserver and intraobserver reproducibility than the KT-1000 [16,31]. Studies have also reported that the GNRB has higher sensitivity and reproducibility than the Telos arthrometer [17,32]. The GNRB’s superior measurement might be due to some technological advantages, including pressure control of the patella, control over calf and hamstring activity, and a more accurate transducer.

Although the GNRB has had satisfactory outcomes, the few studies that have compared multiple arthrometers have had inconclusive results. James et al. conducted a systemic review of 16 knee laxity devices to evaluate the most accurate and reliable tool for stress radiography of knee laxity. However, variable results meant that specific recommendations concerning the best technique could not be made [33]. Murgier et al. conducted a more specific comparative study that compared knee laxity with four different arthrometers for ACL tears. They noted average comparability for the KT1000 and GNRB and poor comparability for the others. Hence, they concluded that the devices were not comparable [25].

Another unresolved problem for arthrometers is what traction forces lead to the most accurate measurement. The international reference force for assessing the ACL with the KT-1000 is 134 N; thus, 134 N was also used as the reference in the GNRB guidebook. However, Lefevre et al. argued that the sensitivity under 250 N was superior to that under 134 N, and recommended a difference of 2.5 mm as a threshold of side-to-side anteroposterior laxity [17]. Moreover, some studies have indicated low reliability under 250 N [33,34]. In our experience from this study, most patients experienced pain and discomfort from a traction force of 250 N, and resulting muscle contraction might confound or even terminate the measurement. Therefore, we applied traction forces of 134 N and 200 N in our study, with the side-to-side laxity measuring 0.76 mm and 0.82 mm, respectively. The threshold under the 134 N suggested “no tear” if the difference was lower than 1 mm. Both traction forces achieved the threshold and demonstrated excellent stability. Another slope parameter was SSD, expressed as S2 (μm/N), which also revealed a low risk of functional instability. After anatomic ACL and ALL reconstruction, the objective value was perfectly demonstrated when measured using the GNRB.

Our study had some limitations. First, it was a retrospective cohort study with a small sample size. Second, the lack of a comparison group with isolated ACL reconstruction might have weakened the interpretation of our measurements. Third, the follow-up period was relatively short. An extended evaluation of combined ACL and ALL reconstruction sustainability is required. Fourth, relatively large standard deviations might lead to some bias for interpretation of the results. Finally, we evaluated rotatory stability only with physical examination in the clinic. Therefore, we lacked a definitive measurement. However, measuring rotatory stability with an arthrometer is difficult, and the reproducibility is tenuous. Even if more standardized arthrometer evaluation methods are developed, we assume that the pivot shift test will remain a relatively more reliable method for rotatory evaluation.

## 5. Conclusions

In conclusion, our results indicate that a good prognosis can be achieved with combined reconstruction of the ACL and ALL. Furthermore, excellent outcomes for both side-to-side differential anterior laxity and stability were demonstrated using a GNRB arthrometer.

## Figures and Tables

**Figure 1 medicina-59-00366-f001:**
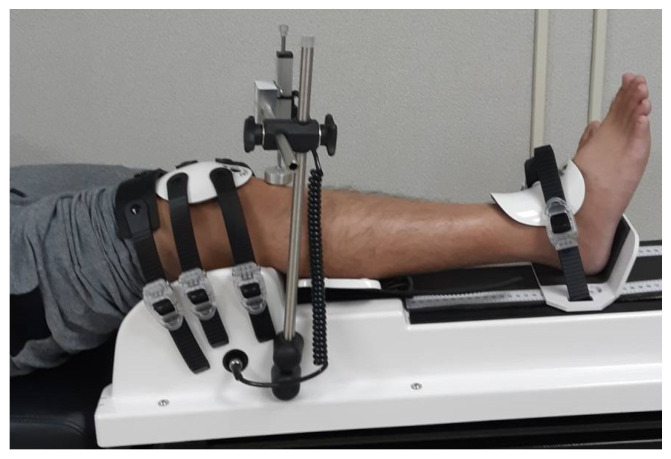
GNRB (Genourob, Laval, France) arthrometer used in our institute.

**Figure 2 medicina-59-00366-f002:**
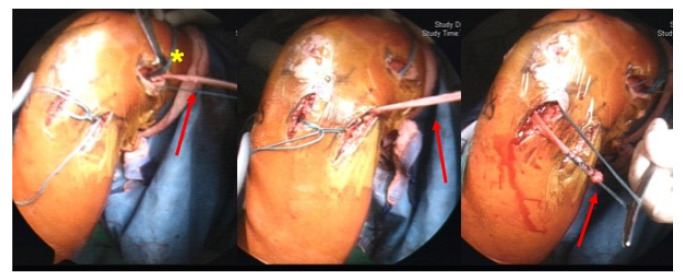
Image of combined ACL and ALL reconstruction. (Red arrow: gracilis was used as an autograft for ALL reconstruction. Asterisk: the anatomic femoral tunnel of the ALL.)

**Figure 3 medicina-59-00366-f003:**
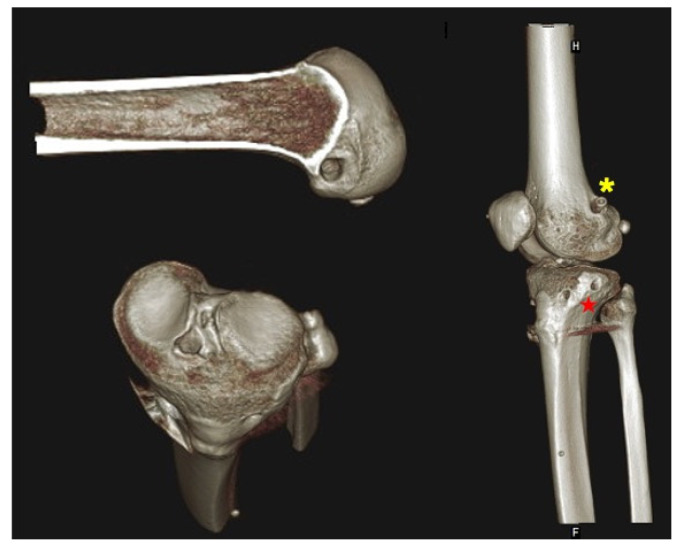
One patient with follow-up computed tomography imaging who received surgery using our technique of combined ACL and ALL reconstruction. The side-to-side laxity under 134 N with the GNRB arthrometer was 0.1 mm for this patient. (Asterisk: the anatomic femoral tunnel of ALL located at the lateral epicondyle posteriorly and proximally. Star: the anatomic tibial tunnel of ALL located between the Gerdy’s tubercle and the fibular head.)

**Table 1 medicina-59-00366-t001:** General data and associated injury.

Sex (male/female)	14/4
Age (years)	30.56 (19–53) +/− 8.9
Time from injury to surgery (months)	22.83 (2–120) +/− 37
Follow up (months)	27.37 (24–32) +/− 3.4
Associated injury	
Lateral meniscus	7
Medial meniscus	3
Medial collateral ligament	2
Lateral collateral ligament	2

**Table 2 medicina-59-00366-t002:** Outcomes of subjective knee evaluations and pivot shift tests.

	Preoperative	Postoperative	*p* Value
IKDC score	54.3 +/− 9.0	85.1 +/− 1.6	<0.001 *
Lysholm score	52.3 +/− 3.3	98.1 +/− 2.2	<0.001 *
Pivot shift (IKDC grade)			
0	0	16	
1	0	2	
2	0	0	
3	18	0	

* A *p*-value of <0.05 indicated significant difference.

**Table 3 medicina-59-00366-t003:** Outcomes of GNRB arthrometer measurement.

	Preoperative	Postoperative	*p* Value
Side-to-side laxity under 134 N (mm)	3.50 +/− 0.80	0.76 +/− 0.78	<0.001 *
Side-to-side laxity under 200 N (mm)	4.38 +/− 1.30	0.82 +/− 0.80	<0.001 *
SSD S2 (um/N)	8.05 +/− 3.50	3.52 +/− 2.17	0.002 *

* A *p*-value of <0.05 indicated significant difference.

## Data Availability

The data underlying this article can be shared on reasonable request.

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
