# Peer review of "Outcomes of Combined Anterior Cruciate Ligament and Anterolateral Ligament Reconstruction According to GNRB Arthrometer Measurement"

_medicina, 2023, doi:10.3390/medicina59020366_

Round 1

Reviewer 1 Report

The article title"Outcomes of combined anterior cruciate ligament and anterolateral ligament reconstruction according to GNRB arthrometer measurement". The aim of this retrospective study was to investigate the prognosis of combined anterior cruciate ligament (ACL) and anterolateral ligament (ALL) reconstruction, we used a GNRB (Genourob, Laval, France) arthrometer to measure the surgical outcomes. Conclusion of this study revealed that combined ACL and ALL reconstruction has an excellent prognosis.

Author Response

Point 1:

The article title"Outcomes of combined anterior cruciate ligament and anterolateral ligament reconstruction according to GNRB arthrometer measurement". The aim of this retrospective study was to investigate the prognosis of combined anterior cruciate ligament (ACL) and anterolateral ligament (ALL) reconstruction, we used a GNRB (Genourob, Laval, France) arthrometer to measure the surgical outcomes. Conclusion of this study revealed that combined ACL and ALL reconstruction has an excellent prognosis.

Response 1:

Thank you for providing these insights. We also appreciate the time and effort you and each of the reviewers have dedicated to providing insightful feedback on ways to strengthen our paper.

Reviewer 2 Report

This article presents a novel quantification of knee stability following combined ACL+ALL reconstruction. In order to strengthen the manuscript, I have two primary suggestions.

First, I would encourage the authors to include some comparison data (either from their own work or literature) directly comparing common outcomes from the GNRB system and more common measures of knee laxity.

Second, please include reports on the repeatability of the measurements performed in this study.

Author Response

Point 1:

First, I would encourage the authors to include some comparison data (either from their own work or literature) directly comparing common outcomes from the GNRB system and more common measures of knee laxity.

Response 1:

Thank you for your suggestion. We have added the paragraph of more informations of GNRB arthrometer and the comparative studies with other arthrometers in “introduction” (line 50-63)

Point 2:

Second, please include reports on the repeatability of the measurements performed in this study.

Response 2:

Thank you for the valuable comment. We have added the report of reproducibility of GNRB in “introduction.” (line 60-63)

Reviewer 3 Report

The author aim to publish a retrospective cohort review including patients with combined ACL and ALL reconstruction with surgeries performed at the authors´ institute.

Standard evaluation methods were used for assessing surgery outcome over two years supplemented by an arthrometer called GNRB.

Overall, the paper is well written and easy to understand. Paper sections are appropriate and data is presented clearly.

However, there are some critic points, which unfortunately lead to the recommendation of major revisions. The two main critic points are:

1.       What is the purpose of the study?

To compare surgical outcomes or to introduce/compare an alternative evaluation technique? The paper does not focus on one of the two mentioned topics.

At the end of the introduction section, the author state that they aimed to evaluate the objective outcome of combined ACL and ALL reconstruction by using the institute´s GNRB. Overall, the introduction is rather short. It gives some general information about the roles of the ACL and ALL, presenting numerous references for single statements. The GNRB is only briefly summarized although it seems to be the main focus. There needs to be a part regarding advantages and disadvantages of this technique and why the authors think it would add valuable additional (or better) information.

2.       What is the originality of this study?

According to the introduction, it seems that it is already known that combined ACLR and ALLR leads to superior outcome compared to single reconstruction (page 1, last paragraph). According to the discussion, there are already papers published about the usability of GNRB in orthopaedics in general and for evaluation of the knee laxity in particular (page 7, first paragraph).

It is suggested that the authors rewrite especially the introduction and supplement the discussion with special focus on the use of GNRB (advantages, disadvantages, what is measured in comparison to other techniques, why is objective outcome important). This is not to be understood wrongly, in research objective outcomes are always of key importance; however, in the light of patients´ satisfaction on the one hand and the prognosis of re-ruptures or other complications on the other hand, it could be possible, that subjective measures would also be sufficient, therefore please consider in your discussion.

Additionally to these major points, there are some other problems:

1.       Page 2, paragraph 2.1 study design: There is nowhere written at which time points which kind of evaluation took place. Please include these information for all techniques used.

2.       Page 2, paragraph 2.2 GNRB, lines 81 -182: Why were these values selected, what was the basis for this selection.

3.       Page 3, paragraph 2.2 GNRB, line 88: the value of the figure would be significantly higher with a patient/leg installed in the device, with marks on the mentioned anatomic postitions.

4.       Pages 3 and 4, paragraph 2.4 surgical technique: overall, since the aim of the study was to evaluate the objective outcome of combined ACLR und ALLR by GNRB, the surgical technique is described rather long. Additionally, in figures 2 and 3 any annotations are missing and need to be supplemented.

5.       Pages 4 and 5, paragraph 3.1 general data: in line 144 on page 4, eleven months are given as mean time from injury to surgery while table 1 on page 5 states 22.83 months as mean time from injury to surgery. Please correct where necessary.

Furthermore, there is a huge standard deviation regarding this parameter. Could this influence the outcome? Please add corresponding paragraph in the discussion.

7.       Page 5, paragraph 3.3 objective values of GNRB: line 164: SSD is defined as side-to-side differential. In the included table (table 3) SSD is written in all three lines although – according to the text – it seems that the two upper lines refer to the side-to-side laxity. Please revise.

8.       Page 6, paragraph 4 discussion, line186 – 188: In the discussion, it is not useful to refer to a review without giving a statement of how the results refer to the study presented. In the referenced lines it is only written that there are variation in the anatomic footprint and heterogeneous biomechanical results, and variable outcome of objective measurement. An explanation or discussion is missing which influence the anatomic variety of footprint has on the outcome, in what extend the biomechanical results are heterogeneous and how this might influence the evaluation of the GNRB results or more generally the use of this technique. Furthermore, what to objective measurements refers the review and is there a link to the own data.

Without this discussion, the mentioning of the review is useless (also a reference number is missing).

9.       Page 6, line 190: number of reference is missing (27).

10.   Page 6, line 190: “The femoral tunnel is made the same as the ACL and….”. What does that mean?

11.   Page 7, lines 230 – 232: “However, Lefevre et al. argued that the sensitivity under 250 N was superior to under 134 N and recommended a difference of 2.5 mm as a threshold.” This sentence is difficult to understand. A difference of 2.5 mm of what? What threshold?

Author Response

Point 1:

What is the purpose of the study?

To compare surgical outcomes or to introduce/compare an alternative evaluation technique? The paper does not focus on one of the two mentioned topics.

At the end of the introduction section, the author state that they aimed to evaluate the objective outcome of combined ACL and ALL reconstruction by using the institute´s GNRB. Overall, the introduction is rather short. It gives some general information about the roles of the ACL and ALL, presenting numerous references for single statements. The GNRB is only briefly summarized although it seems to be the main focus. There needs to be a part regarding advantages and disadvantages of this technique and why the authors think it would add valuable additional (or better) information.

Response 1:

Thank you for the suggestions. The purpose of the study is aimed to measure the objective value using anatomically combined ACL and ALL reconstruction. Using the novel GNRB arthrometer, the data could be more precise. We have added the paragraph of previous studies comparing the GNRB to tradittional arthrometer in “introduction” (line 50-63).

Point 2:

What is the originality of this study?

According to the introduction, it seems that it is already known that combined ACLR and ALLR leads to superior outcome compared to single reconstruction (page 1, last paragraph). According to the discussion, there are already papers published about the usability of GNRB in orthopaedics in general and for evaluation of the knee laxity in particular (page 7, first paragraph).

It is suggested that the authors rewrite especially the introduction and supplement the discussion with special focus on the use of GNRB (advantages, disadvantages, what is measured in comparison to other techniques, why is objective outcome important). This is not to be understood wrongly, in research objective outcomes are always of key importance; however, in the light of patients´ satisfaction on the one hand and the prognosis of re-ruptures or other complications on the other hand, it could be possible, that subjective measures would also be sufficient, therefore please consider in your discussion.

Response 2:

Thank you for these valuable comments. The originality of our study is trying to using the novel GNRB arthrometer to evaluate the objective prognosis of combined ACL and ALL reconstruction. We have added the paragraph of more informations of GNRB arthrometer and the comparative studies with other arthrometers in “introduction” (line 50-63). We agree with you that the subjective evaluations were also important for the clinical outcomes and aimed to do further research for the compatibility of objective and subjective ourcomes.

Point 3:

Page 2, paragraph 2.1 study design: There is nowhere written at which time points which kind of evaluation took place. Please include these information for all techniques used.

Response 3:

Thank you for the comment. We have added the time points and the techniques of GNRB measurement. (line 84-86)

Point 4:

Page 2, paragraph 2.2 GNRB, lines 81 -182: Why were these values selected, what was the basis for this selection.

Response 4:

Thank you for these the valuable comments. 134 N is the international reference force for assessing the ACL baesd to the traditional arthrometer -KT1000, using the force to compare the displacement differential between both knees of both patients at this force. 150N and 200N were then tested to provide the the analysis of the slope of the curves which indicate not only the side to side difference but also the functional instability. We had added the supplement in paragraph 2.2 GNRB (line 92-93)

Point 5:

Page 3, paragraph 2.2 GNRB, line 88: the value of the figure would be significantly higher with a patient/leg installed in the device, with marks on the mentioned anatomic postitions.

Response 5:

We agree with your assessment. We have changed the figure to a patient measured the knee laxity with GNRB (figure 1). Thank you.

Point 6:

Pages 3 and 4, paragraph 2.4 surgical technique: overall, since the aim of the study was to evaluate the objective outcome of combined ACLR und ALLR by GNRB, the surgical technique is described rather long. Additionally, in figures 2 and 3 any annotations are missing and need to be supplemented.

Response 6:

Thank you for providing these insights. We have shortened the paragraph of the surgical technique and have added the annotations in figure 2 and 3. (line 108-125; figure 2 and figure 3)

Point 7:

Pages 4 and 5, paragraph 3.1 general data: in line 144 on page 4, eleven months are given as mean time from injury to surgery while table 1 on page 5 states 22.83 months as mean time from injury to surgery. Please correct where necessary.

Furthermore, there is a huge standard deviation regarding this parameter. Could this influence the outcome? Please add corresponding paragraph in the discussion.

Response 7:

Thank you for these the valuable comments. We haved corrected the “mean time from injury to surgery” as 22.83 months. (line 139-140); We aggreed the huge standard deviation might lead to some bias for interpretation of the result. We have added if to the “discussion”. (line 248-249)

Point 8:

Page 5, paragraph 3.3 objective values of GNRB: line 164: SSD is defined as side-to-side differential. In the included table (table 3) SSD is written in all three lines although – according to the text – it seems that the two upper lines refer to the side-to-side laxity. Please revise.

Response 8:

Thank you. We have reviesd the table 3 and correct the two of the“SSD” to “Side to side laxity under 134N and 200N”.

Point 9:

Page 6, paragraph 4 discussion, line186 – 188: In the discussion, it is not useful to refer to a review without giving a statement of how the results refer to the study presented. In the referenced lines it is only written that there are variation in the anatomic footprint and heterogeneous biomechanical results, and variable outcome of objective measurement. An explanation or discussion is missing which influence the anatomic variety of footprint has on the outcome, in what extend the biomechanical results are heterogeneous and how this might influence the evaluation of the GNRB results or more generally the use of this technique. Furthermore, what to objective measurements refers the review and is there a link to the own data.

Without this discussion, the mentioning of the review is useless (also a reference number is missing).

Response 9:

Thank you for your suggestion and we agree with your assessment. We have added the discussion of anatomic variety of footprint (line 184-193). The beneficial of our current study is that the surgical method we used for ACL and ALL reconstruction by Santi’s group had reported well prognosis. Using GNRB arthrometer, the objective outcome could be further demonstrated. The references were added. (reference 7 and 22)

Point 10:

Page 6, line 190: number of reference is missing (27).

Response 10:

Thank you. We have added the reference 27 in the paragragh. (line 189)

Point 11:

Page 6, line 190: “The femoral tunnel is made the same as the ACL and….”. What does that mean?

Response 11:

Thank you for the comment. We had added more discusssion to explain the technique. (line 190-1932)

Point 12:

Page 7, lines 230 – 232: “However, Lefevre et al. argued that the sensitivity under 250 N was superior to under 134 N and recommended a difference of 2.5 mm as a threshold.” This sentence is difficult to understand. A difference of 2.5 mm of what? What threshold?

Response 12:

Thank you for the valuable comment. We have re-write the sentence. (line 231-233)

Round 2

Reviewer 3 Report

Overall, the paper has been changed according to the suggestions and the questions have been answered mostly satisfactorily.